

# Effects of glycopyrronium bromide versus anisodamine hydrobromide in preventing nausea and vomiting and relieving spasm after ERCP: a randomized controlled trial

Qingjing Ma[1,*], Dina Sun[2,*], Yan Rao[1], Yi Yang[1], Jie Liu[1], Chunmu Miao[3], Guangyou Duan[1], Guizhen Chen[1] and Jie Chen[1]

[1] Department of Anesthesiology, The Second Affiliated Hospital, Chongqing University of Medical Science, Chongqing, China
[2] Department of Gynecology and Obstetrics, The Second Affiliated Hospital, Chongqing University of Medical Science, Chongqing, China
[3] Department of Hepatobiliary Surgery, The Second Affiliated Hospital, Chongqing University of Medical Science, Chongqing, China
[*] These authors contributed equally to this work.

Corresponding authors
Guizhen Chen, 305450@hospital.cqmu.edu.cn
Jie Chen, 305045@hospital.cqmu.edu.cn

## ABSTRACT

**Background.** Postoperative nausea and vomiting (PONV) frequently occur in patients undergoing endoscopic retrograde cholangiopancreatography (ERCP) under general anesthesia. Glycopyrronium bromide, an anticholinergic medication, is believed to not only relieve gastrointestinal spasms but also effectively prevent PONV.

**Objective.** To compare the incidence of nausea and vomiting and the effect of relieving spasm between patients administered glycopyrronium bromide and those given anisodamine hydrobromide following endoscopic retrograde cholangiopancreatography (ERCP).

**Design.** This is a monocentric prospective study.

**Methods.** Patients eligible for ERCP were randomly assigned to two groups. One group received 0.2 mg glycopyrronium bromide (Group G) intravenously, while the other group received 10 mg anisodamine hydrobromide (Group A) intramuscularly for anesthesia induction. The study assessed duodenal motility during ERCP and the incidence of PONV within 24 hours.

**Results.** The study included 130 patients. Nausea and vomiting within 24 hours post-surgery occurred in nine patients (13.8%) in Group G and 19 patients (29.2%) in Group A, with statistical significance (relative risk (RR), 0.47; 95% confidence interval (CI) [0.02–0.29]; $p = 0.033$). Vomiting specifically was observed in three patients (4.6%) in Group G and 12 patients (18.5%) in Group A, showing statistical significance (RR 0.25; 95% CI [0.03–0.25]; $p = 0.028$). There was no significant difference in duodenal peristalsis between the groups (Group G: $10.9 \pm 3.1$ times/min; Group A: $11.6 \pm 3.1$ times/min; $p = 0.174$).

**Conclusion.** For patients undergoing ERCP under general anesthesia, a subcutaneous injection of 0.2 mg glycopyrronium bromide significantly reduces PONV and provides similar anti-spasmodic effects to 10 mg intramuscular anisodamine hydrobromide.

## INTRODUCTION

Endoscopic retrograde cholangiopancreatography (ERCP), is the primary treatment for pancreatic and biliary diseases, offering a direct, effective and minimally invasive approach (*Pekgöz, 2019*). General anesthesia is widely used for complex and painful interventional ERCP procedures (*Raymondos et al., 2002*; *Azimaraghi et al., 2023*). However, postoperative nausea and vomiting (PONV) are common and often unavoidable complications of general anesthesia (*Rajan & Joshi, 2021*). The incidence of nausea is approximately 50%, vomiting occurs in about 30% of cases, and in high-risk groups, the incidence of PONV can reach up to 80% (*Kienbaum et al., 2022*; *Koivuranta et al., 1997*; *Chen et al., 2020*).

Several factors can increase the risk of PONV in patients undergoing ERCP, including use of general anesthetics (especially inhalation anesthetics), opioids, pneumoperitoneum caused by endoscopic surgery, female sex, nonsmoking status, a history of PONV or motion sickness, and age under 50 years (*Kienbaum et al., 2022*; *Gan et al., 2014*). Despite limited prospective studies on nausea and vomiting following ERCP, PONV remains a significant clinical concern in patients receiving general anesthesia. Medications commonly used for PONV prevention include ondansetron, tropisetron, metoclopramide, dexamethasone, fluphenazine (*Weibel et al., 2020*; *Tateosian, Champagne & Gan, 2018*). These medications work through various mechanisms, leading to diverse clinical approaches to managing PONV (*Gan et al., 2014*). However, the incidence of PONV remains high (*Stasiowski et al., 2022*; *Yi et al., 2022*; *Riemer et al., 2021*). Thus, new anti-PONV medications with different mechanisms are needed to provide additional options for improving PONV management.

As reported in one recent paper and one review, it is recommend that anisodamine can be used to relieve gastrointestinal spasms (*Zhang et al., 2023*; *Xia et al., 2024*). And furthermore, expert consensus also suggested that anisodamine can be used for antispasmodic treatment during ERCP (*Yu et al., 2018*). Thus, in this study anisodamine was considered as control treatment in patients receiving ERCP. However, the incidence of nausea and vomiting during the ERCP procedure remains high even after the use of anisodamine. Nausea and vomiting are issues that urgently need to be addressed. In this study we hypothesized that glycopyrronium bromide can not only be used for spasmolysis but also effectively prevent PONV. Unlike non-selective anticholinergics that target M1, M2, and M3 receptors, glycopyrronium bromide is a long-acting quaternary anticholinergic medication and a selective M receptor antagonist with a high affinity for M1 and M3 receptors. Its primary effects include reducing gland secretion and smooth muscle contraction (*Blair, 2021*). This can decrease the occurrence of vomiting by inhibiting gastrointestinal smooth muscle and polyoric contraction (*Wood, 2007*). Additionally, some studies suggest that glycopyrronium bromide may reduce nausea and vomiting during the perioperative period, although these studies did not focus on nausea and vomiting (*Jain & Sharma, 2015*; *Marples & Wrench, 1999*). Based on the

mechanism of action of glycopyrronium bromide and existing studies, we hypothesize that glycopyrronium bromide, as a safe and effective spasmolytic agent, can also reduce the incidence of postoperative nausea and vomiting and promote postoperative recovery. Therefore, this study aimed to compare the effects of glycopyrronium bromide and anisodamine hydrobromide in preventing nausea and vomiting after ERCP and to explore the antispasmodic effects of glycopyrronium bromide and its benefits in reducing the incidence of PONV.

## METHODS

### Study design

This is a monocentric prospective study. The reporting of this study complies with the guideline of Consolidated Standards of Reporting Trials (CONSORT) (*Schulz, Altman & Moher, 2010*). This study was a double-blind, randomized controlled trial, approved by the Ethics Committee of the Second Affiliated Hospital of Chongqing Medical University (Approval ID: 2023-115). Data collection occurred from October 1, 2023, to July 31, 2024, spanning 10 months. This study was registered with the clinicaltrials.gov Protocol Registration and Results System (PRS) (ID: NCT06045364). Informed consent was obtained from all participants.

### Patients

We conducted a double-blind randomized trial involving 130 inpatients at our center who met the criteria for eligibility including aged 18–80, ASA grade I–III from the American Society of Anesthesiologists, with preoperative diagnosis of biliary tract obstruction or pancreatic space - occupying lesions who are able to provide informed consent and who are scheduled to undergo resection or examination during ERCP. Patients were excluded if they had hypersensitivity to anticholinergic drugs, were pregnant, had glaucoma, myasthenia gravis, obstructive gastrointestinal diseases (such as gastric outlet obstruction, intestinal obstruction, or achalasia), obstructive urinary tract diseases (such as benign prostatic hyperplasia), cardiovascular diseases (including coronary heart disease or congestive heart failure), hyperthyroidism, a history of ERCP surgery, chronic renal failure or they declined to participate in the study. All patients provided written informed consent before participation. The full protocol and anonymized data are available upon reasonable request from the corresponding author.

### Randomization, masking and allocation concealment

Patients were randomly assigned to two groups: one received glycopyrronium bromide (Group G) and the other received anisodamine hydrobromide (Group A). Simple randomization, based on computer-generated random numbers, was implemented using the sealed envelope method. An independent investigator assigned patients to groups and sealed the envelops before the end of surgery, storing them at the investigation site until the study's conclusion. All patients, procedural doctors, and researchers were blinded to the group allocation. Anesthesiologists responsible for drug administration were not involved in data collection, input, and analysis.

## Anesthetic procedure

Routine preoperative checkups were performed the day before surgery. Patients completed informed consent forms after receiving information about the study's protocols and were instructed on how to record nausea and vomiting and assess nausea intensity using the numeric rating scale (NRS). In addition to conventional anesthesia induction, all patients received a dexamethasone injection (10 mg). Dexamethasone was used because it had anti-inflammatory effects and could prevent nausea and vomiting. The use of multiple antiemetic drugs during the study period was more in line with ethical principles. According to group allocation, patients in group G received an intravenous injection of glycopyrronium (0.2 mg), while those in Group A received an intramuscular injection of anisodamine hydrobromide (10 mg). Both of these two drugs were administered after the completion of the anesthesia induction and the disappearance of the patients' consciousness to ensure that the patients were unaware of the treatment they had received. After the operation, follow-up is conducted by a designated anesthesiologist who did not participate in the anesthesia and surgical procedure. The dosages are all the usual dosages as per the instructions on the label.

## Endoscopic retrograde cholangiopancreatography

The all ERCP procedures was performed by the same experienced endoscopist who had conducted over 1,000 ERCP procedures. Duodenal movement around the papilla was recorded on videotape during ERCP in all patients.

## Assessment of the effect of glycopyrronium and anisodamine hydrobromide on duodenal motility

Duodenal motility was assessed based on the mean number of contractions and endoscopists' subjective motility scores. Contractions were recorded on video. Subjective motility scores were categorized as follows: grade 0 (none), grade 1 (mild), grade 2 (severe). The total duration of the procedure, defined as the time from insertion to removal of the scope, was measured each procedure.

## Outcome measurement

The primary outcome was the incidence of PONV, recorded 24 ± 1 h after surgery. Regardless of nausea occurrence, the intensity of nausea (measured by the NRS, 0–10, where 0 represents no discomfort and 10 represents severe discomfort), occurrence of vomiting, and need for intervention for nausea and vomiting within 24 h post-surgery were recorded. Secondary outcomes included duodenal motility, operation duration, incidence of other medication side effects (such as dry mouth, urinary retention, and blurred vision), length of stay, and any additional interventions. Patients with an NRS score of ≥4 or vomiting were classified as experiencing severe nausea.

## Statistical analysis

Based on preliminary observations, the incidence of PONV with anisodamine hydrobromide was 22%, and that in patients with glycopyrronium bromide was 5%. The study was designed as 1:1 parallel controlled difference test, with statistical parameters

set as $\alpha = 0.05$ and $1-\beta = 0.90$. Using PASS sample size calculation software 11.0 (NCSS, LLC, Kaysville, UT, USA), the minimum required sample size per group was determined to be 58. Accounting for a 10% dropout rate, 65 patients per group were needed, totaling 130 patients to ensure adequate sample size. Statistical analysis was performed using SPSS software (version 26.0; SPSS, IBM Corp., Armonk, NY, USA). Continuous data are expressed as mean $\pm$ SD. The $\chi 2$ test (or Fisher's exact test where appropriate) was used for categorical data comparison. Incidences of PONV and other outcomes between group G and group A were compared using Chi-square test or Fisher exact test, and relative risk (RR) with 95% confidence interval (CI) was calculated. The continuous data between two groups were compared using independent sample $t$ test. Multivariate logistic regression analysis was performed to explore the role of grouping factor (Group G and Group A) in occurrence of PONV. In addition, we collected the specific occurrence time of PONV, and Kaplan–Meier analysis was performed to explore the cumulative incidence of postoperative nausea and vomiting (PONV) within 24 h between Group G and Group A. Statistical significance was set at $P < 0.05$.

## RESULTS

As illustrated in Fig. 1, 130 patients were randomized, and none were lost to follow-up. Therefore, 130 patients were included in the final analysis (65 in Group G and 65 in Group A). The demographic and baseline parameters of the patients in the two groups are presented in Table 1. There were no differences in age, sex, American Society of Anesthesiologists (ASA) classification, height, weight, body mass index (BMI), surgery type, endoscopic nasobiliary drainage tube (ENBD), Apfel score, or duration of anesthesia between the two groups. However, significant differences were observed in operation time and amount of bleeding.

As shown in Table 2, the primary outcome of the study, the incidence of PONV (13.8% *vs.* 29.2%, $P = 0.033$) and vomiting (4.6% *vs.* 18.5%, $P = 0.028$) in Group G, was significantly lower than those in Group A. The incidence of significant nausea (4.6% *vs.* 20.0%, $P = 0.016$) was also significantly lower than in Group A. There was no significant difference in the additional intervention for PONV between the two groups (1.5% *vs.* 7.7%, $P = 0.210$).

As shown in Table 3, there were no significant differences in total duration of ERCP (1.2 $\pm$ 1.0 *vs* 1.1 $\pm$ 0.6, $P > 0.05$), the mean number of contractions (10.9 $\pm$ 3.1, 11.6 $\pm$ 3.1, $P > 0.05$), motility scores ($P > 0.05$), mouth dryness (32 (49.2%) *vs.* 42 (64.6%), $P > 0.05$), urinary retention (0 *vs* 0, $P > 0.05$), blurred vision (6 (1.5%) *vs.* 3 (4.6%), $P > 0.05$).

Multivariate logistic regression analysis of PONV revealed that a surgery duration longer than 2 h was a risk factor for PONV (OR = 6.135, 95% CI [1.708–22.044], $P < 0.01$) (Fig. 2A). Glycopyrronium bromide was identified as a protective factor against significant nausea (OR = 5.969, 95% CI [1.477–24.128], $P < 0.05$) (Fig. 2B). and vomiting (OR = 5.134, 95% CI [1.271–20.729], $P < 0.05$) (Fig. 2C).

The survival analysis showed a significant difference in the cumulative incidence of postoperative nausea and vomiting (PONV) within 24 h between Group G and Group A

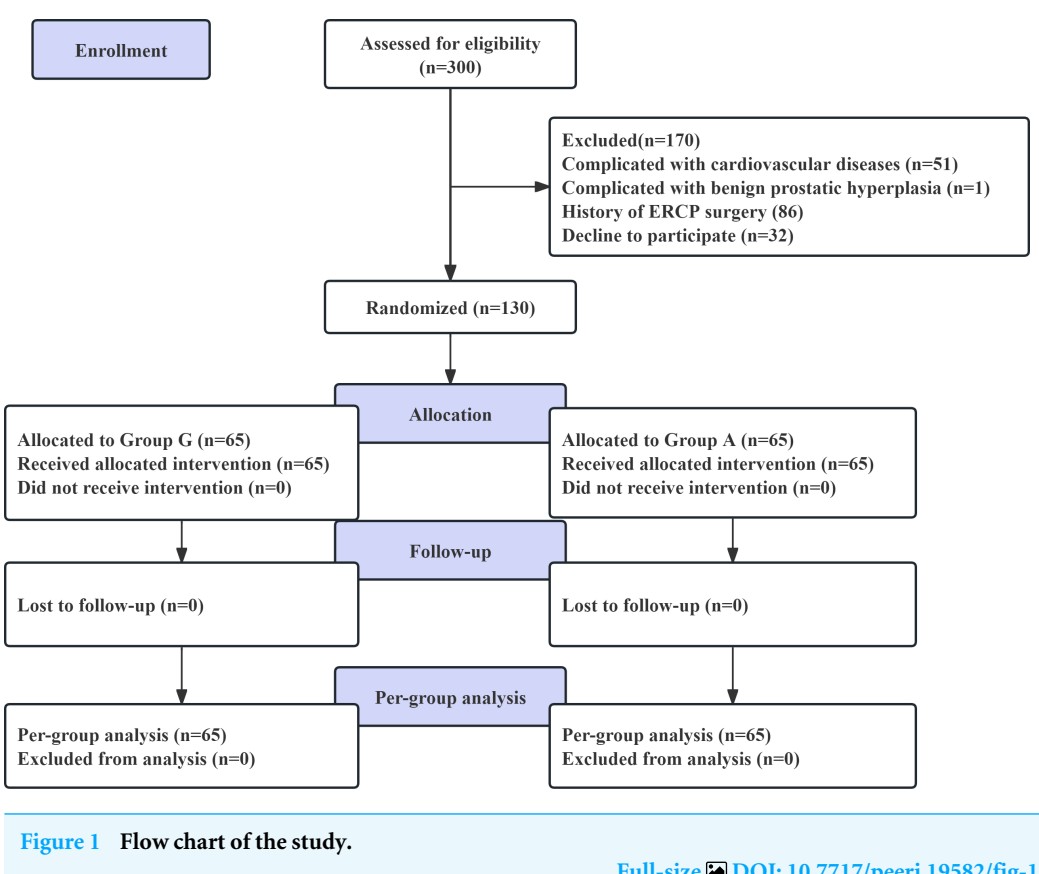

**Figure 1  Flow chart of the study.**

(log-rank $P < 0.05$). Group G significantly reduced the occurrence of PONV, displaying a lower cumulative incidence at each time point (HR < 1), particularly within the first 6 h after surgery (Fig. 3).

## DISCUSSION

In this randomized controlled study, glycopyrronium bromide was found to significantly reduce the incidence of PONV during ERCP, with its spasmolytic effect comparable to that of anisodamine hydrobromide. There are many factors influencing nausea and vomiting, including patient factors, anesthesia factors and surgical factors. In our study, there were no differences among several common influencing factors of PONV. Meanwhile, we performed a multivariate logistic regression analysis on these common influencing factors. The results showed that a surgical duration of more than 2 h was a risk factor for PONV, and glycopyrronium bromide was identified as a protective factor against significant nausea and vomiting. And although the use of dexamethasone may reduce the overall incidence of postoperative nausea and vomiting, dexamethasone was administered to patients in both groups according to the routine method in this study. In clinical practice or studies, a combination of multiple antiemetic drugs was often inevitable. Therefore, we believed that it did not affect the comparison of the effects between glycopyrronium bromide and anisodamine hydrobromide.

**Table 1 Baseline and intraoperative data of the included patients in the two groups.**

| Variable | Group G ($n = 65$) | Group A ($n = 65$) | P value |
|---|---|---|---|
| Age (X ± S, yr) | 54.1 ± 14.4 | 53.4 ± 15.8 | 0.786 |
| Sex ($n$ (%)) | | | 0.219 |
| Male | 36 (55.4%) | 29 (44.6%) | |
| Female | 29 (44.6%) | 36 (55.4%) | |
| Principal diagnosis ($n$ (%)) | | | 1.000 |
| Biliary tract obstruction | 62 (95.4) | 61 (93.8) | |
| Pancreatic space—occupying lesions | 3 (4.6) | 4 (6.2) | |
| ASA classification ($n$ (%)) | | | 0.604 |
| I | 0 | 1 (1.5%) | |
| II | 51 (78.5%) | 50 (76.9%) | |
| III | 14 (21.5%) | 14 (21.5%) | |
| Height, (X ± S, cm) | 162.2 ± 7.4 | 161.0 ± 7.1 | 0.336 |
| Weight, (X ± S, kg) | 64.0 ± 10.6 | 62.3 ± 10.9 | 0.367 |
| BMI (X ± S, kg/m$^2$) | 24.2 ± 3.2 | 24.0 ± 3.7 | 0.682 |
| Surgery type ($n$ (%)) | | | 0.157 |
| ERCP | 32 (49.2) | 24 (36.9) | |
| ERCP+LC | 33 (50.8) | 41 (63.1) | |
| ENBD ($n$ (%)) | 47 (72.3) | 53 (81.5) | 0.212 |
| Operation time, (h) | 2.0 ± 1.5 | 1.8 ± 0.8 | 0.305 |
| Duration of anesthesia, (h) | 2.4 ± 1.2 | 2.4 ± 0.9 | 0.885 |
| Amount of bleeding, (ml) | 12.3 ± 19.3 | 9.7 ± 9.8 | 0.336 |
| Apfel score ($n$ (%)) | | | 0.142 |
| 0 | 8 (12.3) | 2 (3.1) | |
| 1 | 29 (44.6) | 27 (41.5) | |
| 2 | 0 | 1 (1.5) | |
| 3 | 28 (43.1) | 35 (53.8) | |
| 4 | 0 | 0 | |
| Propofol (mg) | 318.8 ± 138.1 | 316.2 ± 116.4 | 0.907 |
| Sufentanil (ug) | 33.5 ± 8.9 | 33.4 ± 8.7 | 0.921 |
| Remimazolam (mg) | 6.9 ± 2.4 | 7.6 ± 4.3 | 0.233 |
| Rocuronium bromide (mg) | 63.8 ± 17.9 | 64.2 ± 18.1 | 0.913 |
| Cisatracurium (mg) | 11.1 ± 2.0 | 13.4 ± 6.7 | 0.403 |
| Remifentanil (mg) | 1.2 ± 0.5 | 1.1 ± 0.5 | 0.345 |

**Notes.**
Abbreviations: ASA, American Society of Anesthesiologists; BMI, Body mass index; ERCP, Endoscopic retrograde cholangiopancreatography; LC, Laparoscopic cholecystectomy; ENBD, endoscopic nasobiliary drainage tube.
The Apfel score includes four risk factors: female sex, nonsmoking status, history of postoperative nausea and vomiting (PONV) or motion sickness, and use of postoperative opioids.

Muscarinic acetylcholine receptors mediate various physiological functions, and five receptor subtypes (M1–M5) have now been identified (*Ishii & Kurachi, 2006*). M1 receptors are primary found in sympathetic postganglionic nerves and gastric wall cells, where they induce excitation and gastric acid secretion. M2 receptors are predominantly located in myocardium, where they influence cardiac contractility and heart rate. M3 receptors are predominantly distributed in the nerve endings of smooth muscle, glands and internal

**Table 2   Postoperative parameters related to nausea and vomiting in two groups.**

| Variable | Group G (n = 65) | Group A (n = 65) | RR (95% CI) | P values |
|---|---|---|---|---|
| PONV (yes, %) | 9 (13.8%) | 19 (29.2%) | 0.47 (0.02 to 0.29) | 0.033 |
| Significant nausea (yes, %) | 3 (4.6%) | 13 (20%) | 0.23 (0.04 to 0.26) | 0.016 |
| Vomiting (yes, %) | 3 (4.6%) | 12 (18.5%) | 0.25 (0.03 to 0.25) | 0.028 |
| Extra intervention for PONV (yes, %) | 1 (1.5%) | 5 (7.7%) | 0.13 (−0.01 to 0.13) | 0.210 |

Notes.
Abbreviations: PONV, postoperative nausea and vomiting; RR, relative risk; CI, confidence interval.

**Table 3   Secondary outcome parameters in two groups.**

| Variable | Group G (n = 65) | Group A (n = 65) | P values |
|---|---|---|---|
| The total duration of ERCP (h) | 1.2 ± 1.0 | 1.1 ± 0.6 | 0.373 |
| Number of contractions (times/min) | 10.9 ± 3.1 | 11.6 ± 3.1 | 0.174 |
| The motility scores (n (%)) | | | 0.851 |
| 0 | 19 (29.2) | 17 (26.2) | |
| 1 | 44 (67.7) | 45 (69.2) | |
| 2 | 2 (3.1) | 3 (4.6) | |
| Mouth dryness (n (%)) | 32 (49.2) | 42 (64.6) | 0.077 |
| Urinary retention (n (%)) | 0 | 0 | 1.000 |
| Blurred vision (n (%)) | 6 (1.5) | 3 (4.6) | 0.300 |

Notes.
Abbreviations: ERCP, Endoscopic Retrograde Cholangiopancreatography.

organs, regulating vascular smooth muscle contraction, bronchial constriction, digestive tract movement, and glandular secretion, including that of the salivary glands, sweat glands, and gastrointestinal tract. They may also influence secretion and movement in the digestive and urinary systems. The pharmacological characteristics of M4 and M5 receptors are less well understood (*Caulfield & Birdsall, 1998*; *Eglen, 2006*). M4 receptors are known to be localized to major visual areas (such as areas V1,17, occipital cortex) and the basal ganglia (*Ferrari-Dileo et al., 1994*); however, characterization of the M5 receptor remains incomplete. Although glycopyrronium bromide and anisodamine hydrobromide are both antichololine drugs, their mechanisms of action differ.

Glycopyrronium bromide is a selective muscarinic receptor antagonist with a higher selectivity for M1 and M3 receptors, being 3–5 times more selective for these receptors than for M2 receptors, resulting in a pronounced peripheral anticholinergic effect (*Chabicovsky et al., 2019*; *Schittek et al., 2021*). Glycopyrronium bromide inhibits smooth muscle contraction and reduces secretions from exocrine glands (such as saliva and bronchial secretions) by competitively antagonizing the action of acetylcholine at muscarinic M receptors, predominantly the M3 subtype. The drug exhibits high selectivity for M3 receptors, which allows it to reduce bronchial secretions and gastrointestinal motility while minimizing effects on the heart (M2 receptors), thereby decreasing side effects such as tachycardia. The half-life of this drug is approximately 33–66 min, with a duration of action of up to 6 h (*Chabicovsky et al., 2019*). Therefore, it has a significant inhibitory effect

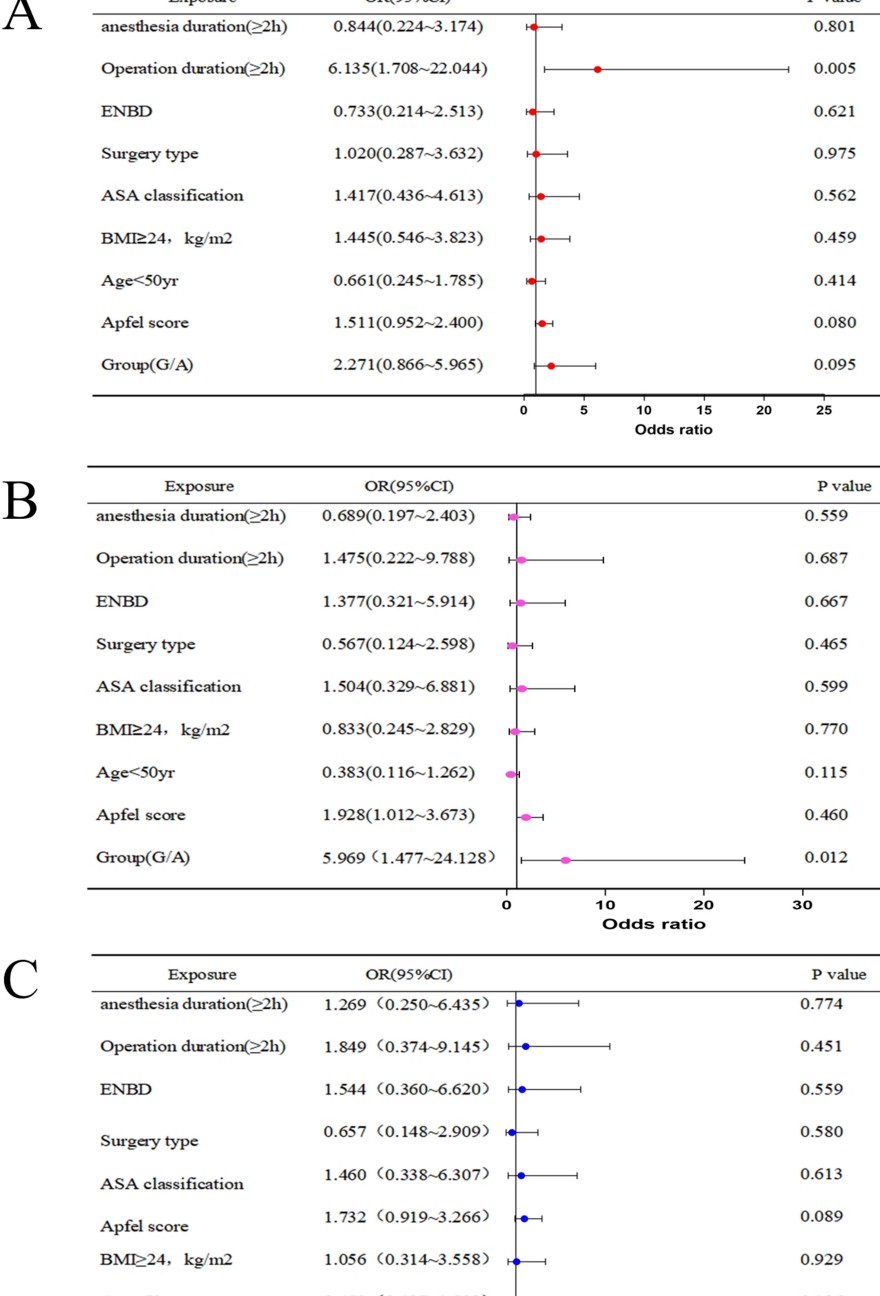

**Figure 2** **Multivariate logistic regression of PONV (A), significant nausea (B), vomiting (C).** Abbreviations: ENBD, endoscopic nasobiliary drainage tube; ASA, American Society of Anesthesiologists; BMI, body max index.

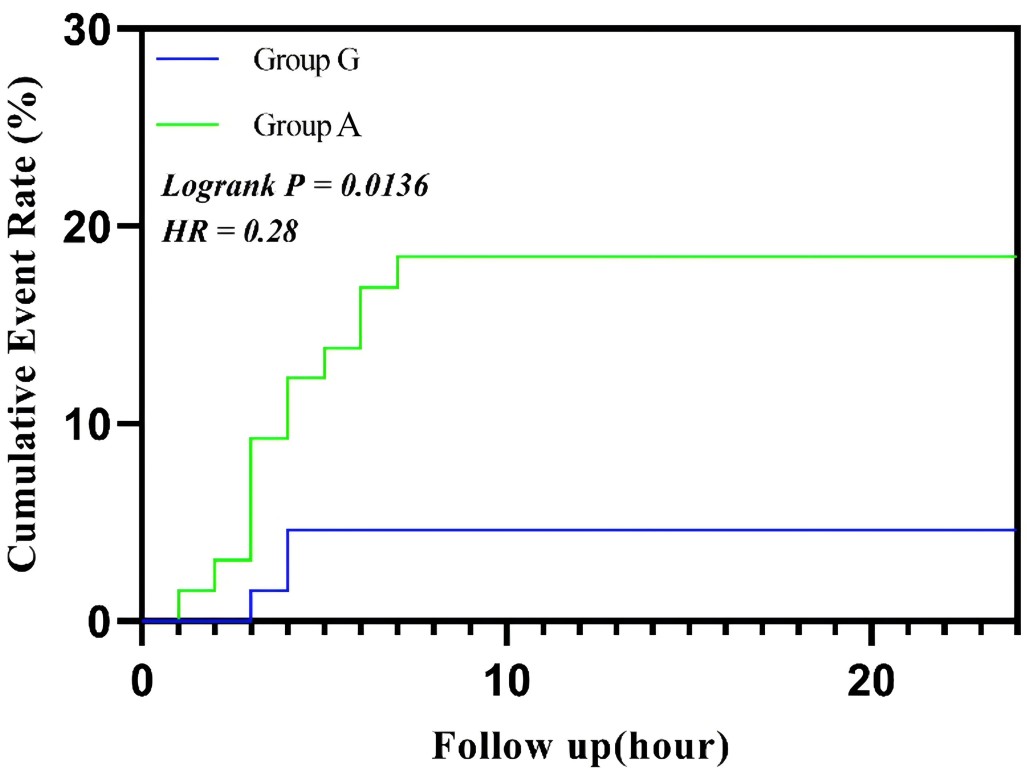

**Figure 3  Cumulative event rate of vomiting.** Abbreviations: HR, hazard ratio.

on gastric acid secretion and gastrointestinal spasm (*Riemer et al., 2021*). Given the high incidence of PONV within 6 h post-surgery, glycopyrronium bromide's duration of action makes it effective in reducing PONV. The primary peripheral anticholinergic effects of glycopyrronium bromide also mean that it has a stronger impact on vomiting than on nausea, consistent with our findings.

Anisodamine hydrobromide, the hydrobromide salt of the natural alkaloid anisodamine, is a peripheral muscarinic receptor antagonist with antispasmodic and microcirculation-improving effects (*Poupko, Baskin & Moore, 2007*). Its mechanism of action is similar to that of scopolamine and atropine, but it has lower central penetrability and stronger peripheral anticholinergic effects. It is commonly used for the treatment of gastrointestinal spasm, septic shock, and other conditions. Scopolamine, when used as an antiemetic, can produce effects similar to antihistamines in preventing PONV (*Furyk, Meek & McKenzie, 2014*) by acting on N2 receptors, antagonizing N1 and M receptors, relieving vasospasm in the ear, improving blood supply, and promoting lymphatic drainage in the labyrinth to reduce edema and relieve vertigo (*Apfel et al., 2010*; *Nachum, Shupak & Gordon, 2006*). Compared to scopolamine, anisodamine hydrobromide has a lower ability to cross the blood-CSF barrier, resulting in fewer central nervous system (CNS) side effects (*Poupko, Baskin & Moore, 2007*). Some studies have reported that ST36 acupoint injection significantly reduce postoperative vomiting in individuals undergoing laparoscopic sleeve gastrectomy, particularly those with obesity. However, this technique did not reduce

postoperative nausea compared to standard prophylaxis (*Xue et al., 2023*), possibly due to the short duration of action and weaker central effects. And the half-life of anisodamine hydrobromide is approximately 40 min, and its duration of action is about 4 h, which is relatively short. This may affect its preventive effect on PONV.

For glycopyrronium bromide, studies have shown that intravenous injection of 0.2–0.4 mg before subarachnoid block in cesarean section and elderly patients effectively prevents intraoperative nausea and vomiting (*Ure et al., 1999*; *Ngan Kee et al., 2013*). Additionally, the anti-PONV effect of glycopyrronium bromide has been found to be comparable to that of ondansetron in subarachnoid block surgery (*Jain & Sharma, 2015*). We hypothesize that glycopyrronium bromide has significant anti-PONV properties, and that its use in patients undergoing ERCP under general anesthesia can reduce the incidence of PONV and provide a spasmolytic effect similar to that of anisodamine hydrobromide. This study confirmed our hypothesis, demonstrating that in patients undergoing ERCP, glycopyrronium bromide's spasmolytics effect was comparable to that of anisodamine hydrobromide, significantly reducing the incidence of PONV, particularly vomiting within the first 6 h post-surgery, thereby benefiting patient prognosis and recovery. A possible mechanism for PONV prevention may involve glycopyrronium bromide selective action on the M1 and M3 receptors, which exerts a peripheral anticholine effect by selectively inhibiting gastric acid secretion and gastrointestinal spasm (*Howard et al., 2017*).

This study had some limitations. First, in this study only patients receiving ERCP were included, the effects of glycopyrronium bromide on other patients required future studies to explore it. Additionally, we only tested the prescribed doses of glycopyrronium bromide and anisodamine hydrobromide, without exploring other dosage. Future research is needed to investigate the clinical effects of different doses according to the patient's weight to determine whether the effect is dose-dependent.

## CONCLUSION

This prospective randomized controlled study demonstrated that, in patients undergoing ERCP surgery under general anesthesia, intravenous administration of 0.2 mg glycopyrronium bromide can significantly reduce PONV. This approach also achieves a comparable spasmolytic effect to a 10 mg intramuscular injection of anisodamine hydrobromide. For patients who require spasmolysis and also need to consider the prevention of PONV, glycopyrronium bromide is a good choice. For patients at high risk of PONV, glycopyrronium bromide is a preferred option. However, the optimal does for glycopyrronium bromide requires further investigation.

### Funding

This work was supported by the Senior Medical Talents Program of Chongqing for Young and Middle-aged. The funders had no role in study design, data collection and analysis, decision to publish, or preparation of the manuscript.

## Grant Disclosures

The following grant information was disclosed by the authors:
The Senior Medical Talents Program of Chongqing for Young and Middle-aged.

## Competing Interests

The authors declare there are no competing interests.

## Author Contributions

- Qingjing Ma conceived and designed the experiments, performed the experiments, analyzed the data, prepared figures and/or tables, and approved the final draft.
- Dina Sun performed the experiments, prepared figures and/or tables, and approved the final draft.
- Yan Rao performed the experiments, authored or reviewed drafts of the article, and approved the final draft.
- Yi Yang performed the experiments, authored or reviewed drafts of the article, and approved the final draft.
- Jie Liu conceived and designed the experiments, authored or reviewed drafts of the article, and approved the final draft.
- Chunmu Miao performed the experiments, authored or reviewed drafts of the article, and approved the final draft.
- Guangyou Duan conceived and designed the experiments, analyzed the data, prepared figures and/or tables, authored or reviewed drafts of the article, and approved the final draft.
- Guizhen Chen conceived and designed the experiments, prepared figures and/or tables, authored or reviewed drafts of the article, and approved the final draft.
- Jie Chen conceived and designed the experiments, analyzed the data, prepared figures and/or tables, authored or reviewed drafts of the article, and approved the final draft.

## Human Ethics

The following information was supplied relating to ethical approvals (i.e., approving body and any reference numbers):

The Ethics Committee of the Second Affiliated Hospital of Chongqing Medical University (Approval ID: 2023-115).

## Data Availability

The raw measurements are available in the Supplementary File.

## Clinical Trial Registration

The following information was supplied regarding Clinical Trial registration:

NCT06045364.

## Supplemental Information

Supplemental information for this article can be found online at http://dx.doi.org/10.7717/peerj.19582#supplemental-information.

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
