# Peer review of "Effects of glycopyrronium bromide versus anisodamine hydrobromide in preventing nausea and vomiting and relieving spasm after ERCP: a randomized controlled trial"

_PeerJ, doi:10.7717/peerj.19582_

## Round 0.1 · original submission · Major Revisions

Please revise according to the reviewer feedback.

Reviewer 1 ·

Basic reporting

The study aims to compare the incidence of nausea and vomiting between patients administered glycopyrronium bromide and those given anisodamine hydrobromide following endoscopic retrograde cholangiopancreatography (ERCP). The title is “Effects of glycopyrronium bromide versus anisodamine hydrobromide in preventing nausea and vomiting and relieving spasm after ERCP: a randomized controlled trial”.
1. This is a monocentric prospective study.
2. Please show the anesthetic drugs used for general anesthesia including type, mean, and total dose of this study.
3. Why did the authors use dexamethasone in the study? This drug might be influencing the outcome of the study.
4. The route of the study drugs is not commonly used in recent clinical practice.
5. Who were the endoscopists? How about their experiences?
6. Anisodamine hydrobromide (Ani HBr) is a Chinese medicine used to improve blood flow in circulatory disorders.
7. Several factors influence the outcome of the study. Please discuss these.
8. Please add more details on the pharmacophysiology of glycopyrronium bromide and anisodamine hydrobromide.
9. Please add more details to the discussion section.
10. What is the new knowledge of the study?
11. Please recommend to the readers “How to apply this knowledge in clinical practice?”.

Experimental design

None

Validity of the findings

None

Additional comments

Poorly written style

Reviewer 2 ·

Basic reporting

This paper described a study comparing the effects of glycopyrronium bromide verse anisodamine hydrobromide for the treatment of PONV in patients who received ERCP. In general, this manuscript is well-written with professional English, with a clear structure and no significant typographical or grammatical errors.

Experimental design

The authors conducted a double-blind randomized trial comparing the investigation drug glycopyrronium bromide versus the control treatment anisodamine hydrobromide. The research question is well-defined and could potentially fill the knowledge gap regarding the effect of glycopyrronium bromide on the treatment of PONV. However, given that there are other medications for the treatment of PONV after ERCP, the rationale of selecting anisodamine hydrobromide as the control treatment should be provided.

Additionally, it’s claimed that the trial is double-blind, however, the drug administration method is different between the two treatment arms and patients may be able to identify the treatment they received (patients in group G received an intravenous injection of glycopyrronium, while those in Group A received an intramuscular injection of anisodamine hydrobromide). Please clarify how can the trial be double-blind.

Finally, some comments on statistical analysis
• Line 136 please clarify the meaning of “with the addition of “? Is the incidence of PONV 5% with glycopyrronium bromide in combination with another drug?
• Line 135-136 please clarify if the alternative hypothesis for the incidence of PONV was 22% for group A and 5% for group G.
• Line 142 please clarify why the Wilcoxon signed-rank test was used
• Line 146 please clarify what an independent sample-size t-test is.
• Line 148-149, please clarify if the time to the occurrence of PONV was studied.

Validity of the findings

There are issues with some results reported. In Table 1, demographic variables are compared between two treatment arms. In Line 160 please clarify if the two-sample t-test or Mann-Whitney U test was used to compare operation time and amount of bleeding between two groups and why the p values were significant. By doing a two-sample t-test with mean and standard deviation provided in the table, the p-values are actually not significant.

In Line 162-164, please check the results presented in the brackets as they are matching with the wrong outcomes.

There were no titles for Figure 2 and it is unclear what figure 2A is for, same with 2B, 2C.

---

## Round 0.2 · accepted · Accept

Congratulation!
With best regards,
Yoshi
Prof. Yoshinori Marunaka, M.D., Ph.D.

Reviewer 2 ·

Basic reporting

No additional comments as all comments from previous review have been addressed.

Experimental design

No additional comments as all comments from previous review have been addressed.

Validity of the findings

No additional comments as all comments from previous review have been addressed.